# Improving the Rapidity of Magnitude Estimation for Earthquake Early Warning Systems for Railways

**DOI:** 10.3390/s24227361

**Published:** 2024-11-18

**Authors:** Shunta Noda, Naoyasu Iwata, Masahiro Korenaga

**Affiliations:** Center for Railway Earthquake Engineering Research, Railway Technical Research Institute, 2-8-38 Hikari-cho, Kokubunji 185-8540, Tokyo, Japan; iwata.naoyasu.19@rtri.or.jp (N.I.); korenaga.masahiro.70@rtri.or.jp (M.K.)

**Keywords:** earthquake early warning, magnitude estimation, time-dependent parameter, high-speed rail

## Abstract

To improve the performance of earthquake early warning (EEW) systems, we propose an approach that utilizes the time-dependence of P-wave displacements to estimate the earthquake magnitude (*M*) based on the relationship between *M* and the displacement. The traditional seismological understanding posits that this relationship achieves statistical significance when the displacement reaches its final peak value, resulting in the adoption of time-constant coefficients. However, considering the potential for earlier establishment of the relationship’s significance than conventionally assumed, we analyze waveforms observed in Japan and determine the intercept in the relationship as a function of time from the P-wave onset. We demonstrate that our approach reduces the underestimation of *M* in the initial P-wave stages compared to the conventional technique. Consequently, we find a significant rise in the number of earlier warnings in the Japanese railway EEW system. Due to the inherent trade-off between the immediacy and accuracy of alarm outputs, the proposed method unavoidably leads to an increase in the frequency of alerts. Nonetheless, if deemed acceptable by system users, our approach can contribute to EEW performance improvement.

## 1. Introduction

Earthquake early warning (EEW) systems can forewarn against strong shaking, allowing humans and systems to undertake action to protect life and property [1,2,3,4,5,6]. EEW systems can be highly effective in the field of railway operation [7]. The Japan Railway (JR) companies operate in-house EEW systems to reduce the risk of the high-speed rail, Shinkansen, during large earthquakes. Running Shinkansen trains are halted by cutting off the power supply when an earthquake warning is issued from a seismometer in the system to the substation [8]. Typically, when an earthquake event is occurring, EEW algorithms, including that of the Shinkansen system, estimate seismic parameters, hypocenter or epicenter location, and magnitude (*M*), and then, a warning is dispatched if it is judged as necessary when examining the source information. Commonly, the strong motion distribution is computed using a ground motion prediction equation (GMPE) as part of this judgment. The Shinkansen EEW algorithm, on the other hand, assesses a potential damage area using the seismic parameters and considering the database of damage to railway facilities in past significant earthquakes (the M-Δ diagram) [8,9,10] to stop the trains. The details of this diagram are described in the Discussion section. Another characteristic of the Shinkansen system is that the single-station method is employed, in which every single station processes the data through observation to produce warning outputs. Despite being generally less accurate than the multiple-station method in terms of issuing the alert, this method offers the advantage of high immediacy [11,12], which is crucial for railway operation.

To estimate *M* for EEW systems, previous studies have proposed an approach based on the frequency content of the initial P-wave (2–3 s after the P-wave’s arrival) [13,14,15,16,17,18,19]. Alternatively, a relationship between *M* and the displacement is useful [17,18,20,21,22,23]. The Shinkansen EEW algorithm utilizes this relationship, and the equation is given by Ref. [23]:*M* = Pm1 × log*Dis* + Pm2 × log*Δ* + Pm3 + Pm4 × *Δ*,(1)
where *Dis* is the displacement, *Δ* is the epicentral distance, and log is the common logarithm. Pm1, Pm2, Pm3, and Pm4 are the coefficients of the equation. In this article, the approach that uses a relationship between *M* and the displacement to estimate *M* is referred to as the displacement approach. In the Shinkansen EEW algorithm, *Δ* is estimated from the C-Δ method that utilizes the significant correlation between *Δ* and the slope of the initial P-wave envelope with a frequency of 10–20 Hz [23]. The reason why the Shinkansen algorithm considers *Δ* rather than the hypocentral distance is that the analysis using initial P-waves based on the C-Δ method does not allow us to infer the source depth. Note that the hypocentral distance may be used instead of *Δ* for the displacement approach in other EEW systems. Pm1 adjusts the correlation between *M* and log*Dis*. Pm2 and Pm4 are associated with the geometric spreading and anelastic attenuation, respectively. Pm3 represents the intercept term of the equation, which is a value that does not depend on other variables.

Equation (1) resembles a GMPE in that the ground motion (in this case, log*Dis*) and *M* terms are typically arranged on the left- and right-hand sides, respectively, to predict the motion intensity. Meanwhile, these are reversed in Equation (1), because the equation is specifically designed to determine *M*. Because the least squares method determines the coefficients by minimizing the residuals of the dependent variable (i.e., the value placed on the left-hand side), the resulting coefficients may not be equivalent if the arrangement of terms changes in the equation.

The displacement approach should be more reasonable for use in EEW systems than employing the frequency content, because *M* can be updated conveniently with the observed displacement as an earthquake becomes larger. That is, the displacement recorded at a station increases with the growth of the rupture. In addition, Ref. [18] demonstrated that the *M* estimation with the displacement approach was more accurate than that from the frequency content.

The aim of this study is to further enhance the rapidity of *M* estimation from the displacement approach in the single-station method utilized in the Shinkansen EEW system and to reduce the time to warning issuance. To achieve this, we apply the technique of Ref. [24], whose authors proposed setting a coefficient in the relationship between *M* and displacement as a function of time. This mitigates the underestimation of *M* in the initial stages of a P-wave. We then evaluate the improvement in the immediacy of warning output and confirm its disadvantages, specifically for the field of railways. Although there is typically a trade-off between speed and accuracy in terms of issuing the alert, the efficacy of the proposed technique remains valuable, provided that the associated drawback is deemed acceptable for the system users.

## 2. Method

The conventional algorithm for determining *M* in EEW systems utilizes the time-constant coefficients for the relationship between *M* and log*Dis*, such as in Equation (1). This implies that the *M* estimate is directly proportional to log*Dis* at a given distance. It is important to note that, from our point of view, the traditional seismological understanding assumed that the statistically significant proportionality between *M* and log*Dis* is established when *Dis* represents the final maximum amplitude. Therefore, the coefficients have been derived through statistical analysis using the final maximum *Dis*. During an ongoing earthquake, in EEW systems, the *Dis* observed up to that point (i.e., the value approaching the final maximum amplitude) must be utilized, even if the earthquake event eventually results in a large *M*. Consequently, the final *M* cannot be obtained until the final maximum *Dis* is observed. Several past studies have indicated that this poses a technical limitation on the rapid determination of *M* in seismology [25,26,27,28].

To consider that limitation, Ref. [29] examined a dataset of accelerograms recorded at stations of Kyoshin Network (K-NET), operated by the National Research Institute for Earth Science and Disaster Resilience (NIED) [30], and found that the significant relationship between the final *M* and log*Dis* was present at earlier stages than conventionally believed. The amplitude continues to increase even after this time point and reaches its final maximum at the previously considered timing. Based on this finding, Ref. [24] proposed a new technique for the displacement approach to improve the estimation performance of *M* in EEW systems. In their proposal, the intercept of the relationship between *M* and log*Dis* varies with time:*M* = Pm1 × log*Dis* + Pm2 × log*Δ* + Pm3[*T*] + Pm4 × *Δ*,(2)
where *T* is the elapsed time from the onset of P. The difference from Equation (1) is that the intercept is a function of time *T* (Pm3[*T*]) in Equation (2). Note that Pm3[*T*] does not represent a specific mathematical expression; rather, it indicates that the value varies over time and is determined by the data. Ref. [24] determined the time-dependent intercept using the least squares method, while keeping the other coefficients fixed. The authors utilized the maximum absolute displacement observed up to each time point, with *T* set to 1.0, 1.25, 1.5, 1.75, 2.0, 2.5, 3.0, and 4.0 s. The time-dependent intercept allows us to adjust the amplitude, which increases even after the significant proportionality between *M* and log*Dis* is established at an early time *T*. In the analysis of Ref. [24], however, *M* and log*Dis* were placed on the right- and left-hand sides, respectively, following the typical form of GMPE. Additionally, the term Pm4 × *Δ* was omitted in their study. We present results obtained using Equation (2), leading to an improvement in the estimation performance of *M*.

## 3. Data

In this study, we use 11,168 accelerograms observed in 1996–2018 at K-NET stations operated by the NIED [30] to derive the coefficients of Equation (2). These were caused by 223 events with 4.5 ≤ *M*_J_ ≤ 8.0 and focal depth ≤ 150 km. *M*_J_ represents the magnitude obtained from the Japan Meteorological Agency (JMA) [31,32]. Smaller or deeper earthquakes, which are less likely to cause damage, are excluded from this study, as the proposed approach is intended for use in EEW systems. Data with epicentral distances (*Δ*) less than 200 km are selected. We deem it necessary to utilize records from such distant stations, as many damaging earthquakes occur offshore in Japan. Figure 1 and Figure 2 present the histograms of *M*_J_ and *Δ*, respectively. To avoid the influence of non-seismic noise when analyzing the data, we visually inspected the waveforms and manually picked P- and S-wave onsets. The sampling frequency of the waveforms used in this study is 100 Hz. The accelerograms are converted into displacements using a causal recursive filter with an infinite impulse response [33]. We compute the running maximum of the absolute vertical displacement and the peak amplitude in the entire P-wave, which is the maximum in the time between the arrival of the P- and S-waves.

## 4. Results

### 4.1. Determination of the Coefficients

Following the approach described in Ref. [24], we determine the coefficients of Equation (2). We employ Pm2 = 0.4535 and Pm4 = 0.003789, which are conventionally used in the Shinkansen EEW algorithm [23]. Using the least squares analysis with *M*_J_ and the peak log*Dis* of the entire P-wave, we obtain a value of Pm1 = 0.9837. The role of Pm1 is to adjust the correlation between the P-wave’s amplitude and magnitude. This correlation starts out weaker immediately after the onset of the P-wave but strengthens over time, reaching its highest level when using the peak amplitude of the entire P-wave. We can improve the accuracy of magnitude estimation by determining and fixing Pm1 from data at the point of strongest correlation [24]. Subsequently, we calculate the intercept Pm3[*T*] using the least squares method at *T* = 1.0, 1.25, 1.5, 1.75, 2.0, 2.5, 3.0, and 4.0 s, with the fixed Pm1 = 0.9837 determined through the running maximum at each time point. The time *T* starts at the manually picked P onset.

Table 1 presents the coefficients of Equation (2) determined in this study and, for comparison, also shows the conventional parameters used in the Shinkansen EEW system [23]. We have obtained Pm3 for the time after *T* = 5 s using the data of the entire P-wave amplitude (that is, Pm3 = 6.1202 has been simultaneously derived with Pm1 = 0.9837). Furthermore, Pm2 and Pm4 are equal between the conventional method and the one proposed in this study, as mentioned above. As shown in the table below, the Pm3[*T*] determined in this study decreases over time. Since *Dis* increases with *T*, decreasing Pm3 enables compensation for this effect.

### 4.2. Accuracy of M Estimates

The accuracies of the *M* estimation methods are compared using the conventional coefficients [23] and the ones proposed in this study. Figure 3 shows the distributions of *M*_J_ and *M*_est_ at each time *T*. Here, *M*_est_ is the estimate of *M* from each equation. The blue and red circles represent the estimates from the conventional and proposed parameters, respectively. The root mean square (RMS) values of the residuals between *M*_J_ and *M*_est_ are shown at the upper left corner of each figure. The solid black lines signify *M*_J_ = *M*_est_. As shown in Figure 3, *M*_est_ is underestimated at early times *T*, particularly for large earthquakes, based on the conventional coefficients. Meanwhile, the underestimation is decreased when the proposed parameters are used. Consequently, the RMS values obtained by the proposed parameters are smaller than those calculated using the conventional ones. For example, the RMSs are 0.942 (Figure 3a) and 0.708 (Figure 3b) at *T* = 1 s when the conventional and proposed coefficients, respectively, are used. This demonstrates that the estimation accuracy is improved by approximately 25% at *T* = 1 s with the new parameter.

Figure 4 shows the time variations of the RMS and indicates those computed using the entire P-wave data on the right-hand side of the figure. The blue, red, and green circles denote the results from the conventional and proposed coefficients, and the estimation technique of Ref. [24], respectively. This demonstrates that the RMS is the lowest until *T* = 4 s when the parameters determined in this study are adopted. Furthermore, there is near equivalence between those from the conventional and proposed coefficients when the entire set of P-wave data are analyzed (the RMSs are 0.411 and 0.405, respectively). The method of Ref. [24] generates a better estimate earlier than the conventional parameter, because the technique of Ref. [24] has a time-dependent intercept. However, the accuracy based on the method of Ref. [24] becomes relatively low over time. We conclude that the coefficients recommended in this study can estimate *M* more accurately and rapidly than conventional techniques, and thereby, the performance of the EEW system can be improved.

## 5. Discussion

### 5.1. Impact on the Rapidity and Frequency of Warnings for Railway Operations

The technique proposed in this study is valuable for reducing the underestimation of *M* at early times, as demonstrated above. Here, we discuss the impact of this technique on the speed and frequency of warnings for railway operations, using the potential damage area derived from a database of past significant earthquakes affecting railway facilities in Japan (the M-Δ diagram) [8,9,10].

Figure 5 illustrates the M-Δ diagram, in which each symbol represents damage resulting from the events indicated in the legend. In this relationship, *M* is directly proportional to the extent of the affected area of damage, as expressed by the following formula:log*Δ* = 0.51 × *M*_J_
*−* 1.5(3)

The lower limit of *M*_J_ is set at 5.5, with an upper limit for *Δ* of 400 km. The yellow-highlighted zone in Figure 5, which encompasses all damages caused by the previous events, depicts the empirically determined damage area based on the past earthquakes, suggesting a potential for damage to railway facilities in future events. Following this idea, the Shinkansen EEW system immediately halts trains that are traveling within the potential damage area as being at risk based on seismic parameters estimated from the EEW algorithm. This is achieved by outputting a warning (halt signal) from its seismometers. Note that being located within the potential damage area corresponds to experiencing seismic intensity of approximately 5-lower or higher on the JMA scale [34,35], according to Ref. [8]. In this study, we consider it necessary for the warning output to be activated that the seismometer estimating seismic parameters is located within the potential damage area (Figure 6). Under this consideration, we aim to verify how the rapidity and frequency of warnings change by introducing the coefficients of the *M* estimation formula proposed in this study compared to using the coefficients that are conventionally employed in the Shinkansen EEW system [23].

For this test, we utilize the same dataset with the coefficient determination (see the Data section). In the actual EEW algorithm, noise discrimination is performed, such as by examining the frequency characteristics of the observed ground motion [23]. Although the original dataset consisted of 11,168 waveforms, the number of accelerograms that can be used for estimating seismic parameters decreases to 9737 in this test through the process of discrimination (this indicates that there is a certain number of cases where the seismic wave is judged to be noise, even though all 11,168 data points were visually confirmed to be earthquakes). In addition, to assess the rapidness of warning, data will be selected from within this set. Here, we extract records observed at stations located within the potential damage area of the M-Δ diagram, derived from seismic information announced by the JMA (that is, in this selection rather than from the parameters estimated based on the EEW algorithm). These data correspond to waveform records for which a warning needed to be issued based on the high-precision source information determined by the agency after the earthquakes and comprises 1217 records (Dataset A). We discuss the changes in alert immediacy due to the introduction of the proposed coefficients using Dataset A.

Figure 7 shows the results indicating the timing of alarm issuance based on the seismic parameters, estimated from the conventional and proposed coefficients by analyzing Dataset A. In other words, this figure indicates the timing at which the station locations have been determined to be within the potential damage area based on the seismic parameters estimated with the EEW algorithm. In the algorithm, the P-wave onsets are picked with the STA/LTA (short-term average/long-term average) method [36], and the epicentral distances are inferred from the C-Δ method [23]. We find that introducing the technique proposed in this study leads to two beneficial effects. The first is that although the conventional method could not issue alerts for a number of cases that required them, the proposed method makes this possible in some instances. As mentioned above, Dataset A corresponds to cases requiring warnings, as it refers to the source information announced by the JMA. However, with the conventional method, it was not possible to issue alerts for 394 cases (no warning: the outcome in blue shown at the left edge of Figure 7). In the proposed method, this number decreased to 311 (the one in red shown at the left edge of Figure 7), indicating a reduction in missed warnings by approximately 21% due to the introduction of the proposed method. The second is the improvement in warning speed. The number of cases for which alerts are issued at 1 s after the P onset increases from 147 to 430, which is about a 2.9-fold increase. When calculating the average timings for cases where warnings are issued between 1 and 4 s from the P-wave start, they are approximately 2.3 s and 1.5 s using the conventional and proposed coefficients, respectively. Additionally, the number of cases that output warnings after 4 s is reduced from 402 to 223. These results demonstrate that adopting the proposed coefficients improves the rapidity of warnings compared to using the conventional parameters.

The proposed method is aimed at reducing the underestimation of *M*, characterized by the trade-off of conducting larger *M* estimations. Although it is difficult in principle to avoid this trade-off relationship, it can still be beneficial to utilize this technique if the impact of overestimation is within the acceptable range for system users. We compute the number of alarm issuance for the dataset of 9737 records in which the seismic parameters are obtained with the EEW algorithm. Consequently, when using the conventional and proposed coefficients, the number of cases in which the seismometers are located inside the potential damage area (i.e., warnings are triggered) based on the seismic parameter estimation are 1314 and 1970, respectively. This indicates that employing the proposed technique results in a frequency of warning outputs that is approximately 1.5 times higher than that of the conventional method.

### 5.2. Improving Shinkansen Safety with Efficient Earthquake Warnings

Even though, fortunately, there have been no fatalities among passengers on Shinkansen trains to date, the 2022 Fukushima earthquake (*M*_J_ 7.4; the origin time = 11:36 PM Japan Standard Time, 16 March 2022) marked the first instance of a train derailment during commercial operation since the 2004 Niigata Chuetsu earthquake (*M*_J_ 6.8; the origin time = 5:56 PM Japan Standard Time, 23 October 2004). Therefore, for railway operators such as the JR companies, halting trains at the earliest opportunity during an earthquake is crucial. The approach proposed in this study is effective for this purpose. As a trade-off, mitigating the rise in warning issuance frequency, which ensues as a consequence, poses a challenge, as shown above. However, at least in the case of Shinkansen operations, there can be a way to address this issue. Shinkansen trains are urgently halted by the seismometer sending a stop warning to the substation and cutting off the power supply [8]. After the earthquake shaking has finished, the distribution of the actual shaking along the rail is quickly confirmed from the data observed by the seismometers of the system deployed along the railway tracks. If the running trains were suspended due to an overestimation of *M*, it would be possible to make a decision on resuming operations based on the distribution of the actual shaking. Then, by restoring the power supply from the substation, trains can be restarted, and this process can take as little as a few minutes. Consequently, we consider that the approach proposed in this study is effective for the Shinkansen EEW system and the operation of Shinkansen trains. In fact, it should be noted that the proposed coefficient is already operational in the EEW system of JR East [37].

In this article, we focused on applying the proposed method within the railway system of Japan and discussed its effectiveness. However, this approach is anticipated to benefit not only railway applications but also EEW systems for the public and in other regions. Moving forward, further studies in this direction are expected to enhance the overall impact of EEW systems. As previously mentioned, the displacement approach allows us to update the *M* estimate in response to changes in amplitude over time. Since the method proposed in this study adjusts the coefficients of the estimation formula used in *M* calculation, it has the advantage of not increasing computational costs, making it more feasible for implementation.

## 6. Conclusions

In this study, we utilize the time-dependence of P-wave displacements in the relationship between *M* and the displacement to determine *M*. Several previous studies suggested that estimating the final *M* was impossible until the final peak displacement was observed, as the correlation of this relationship was only definitively established when *M* and the final maximum amplitude were considered. However, we obtain the coefficient of the relationship in the early stages of initial P-waves by examining the running maximums of displacement recorded at K-NET stations in Japan, as this correlation can be identified earlier than traditionally thought. As a result, the intercept of the relationship is determined to be time-dependent, whereas all of the coefficients were conventionally assumed to be time-constant. Using the proposed parameter, the underestimation of *M* is mitigated during early P-waves. For example, the RMS of the residuals improves by approximately 25% at *T* = 1 s after the P-wave’s arrival due to the reduction in underestimation. When the proposed coefficient is introduced into the Shinkansen EEW algorithm, its advantages over the conventional method can be divided into two types. The first effect is a reduction in missed warnings, with the number decreasing by about 21% when analyzing the dataset used in this study. The second is an improvement in the speed of alert output. We find a significant increase in the number of cases resulting in earlier warnings. Specifically, the number of cases that can issue alerts at *T* = 1 s increases by a factor of approximately 2.9. The implementation of the proposed approach inevitably leads to a higher occurrence of warnings, owing to the inherent trade-off between the immediacy and accuracy of alarm issuance. Our test indicates an increase of about 1.5 times in this regard. Nevertheless, the technique proposed in this paper is expected to improve the performance of the EEW system if this increase is considered acceptable by system operators.

## Figures and Tables

**Figure 1 sensors-24-07361-f001:**
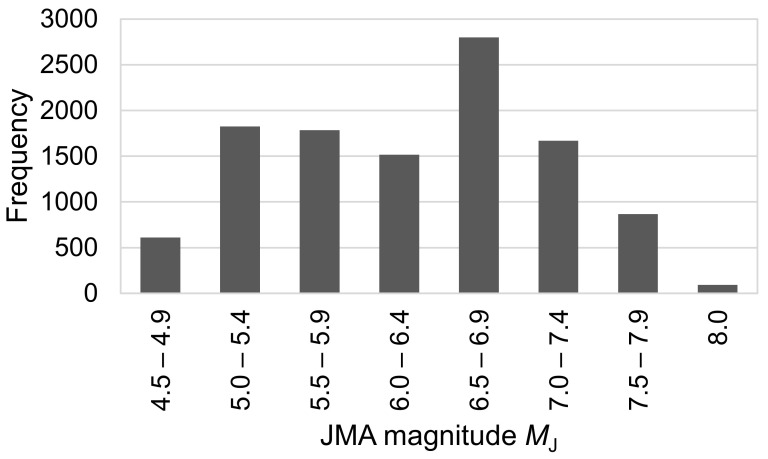
Histogram of the dataset of JMA magnitude, *M*_J_.

**Figure 2 sensors-24-07361-f002:**
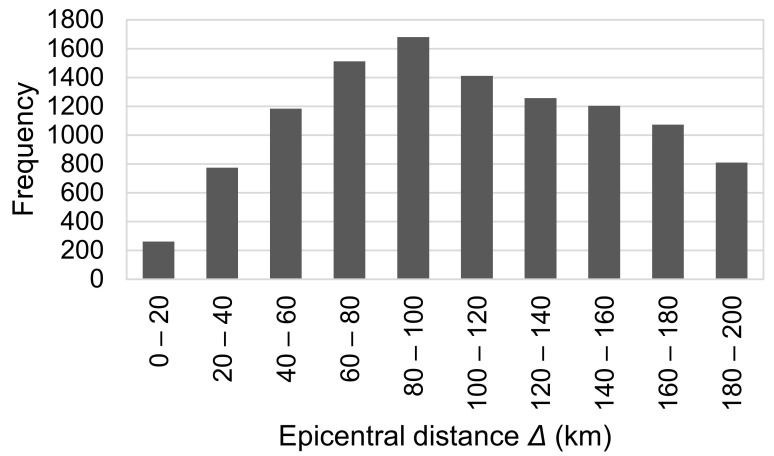
Histogram of the dataset of epicentral distance, *Δ*.

**Figure 3 sensors-24-07361-f003:**
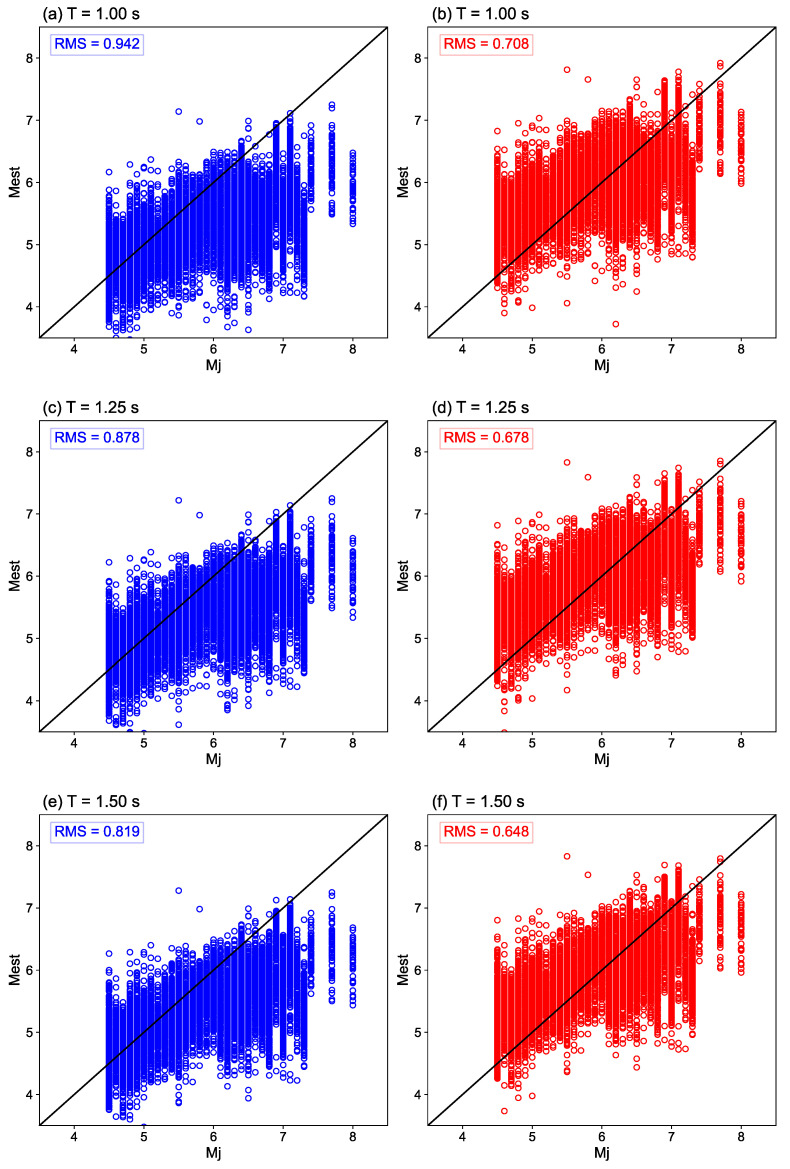
Distribution of *M*_est_ on *M*_J_ at each time *T*. The figures on the left- (blue) and right-hand (red) sides are derived with the conventional and proposed coefficients, respectively. (**a**,**b**) show the results at *T* = 1.0 s, (**c**,**d**) at *T* = 1.25 s, (**e**,**f**) at *T* = 1.5 s, (**g**,**h**) at *T* = 1.75 s, (**i,j**) at *T* = 2.0 s, (**k**,**l**) at *T* = 2.5 s, (**m**,**n**) at *T* = 3.0 s, (**o**,**p**) at *T* = 4.0 s, and (**q**,**r**) are obtained using the entire P-wave data.

**Figure 4 sensors-24-07361-f004:**
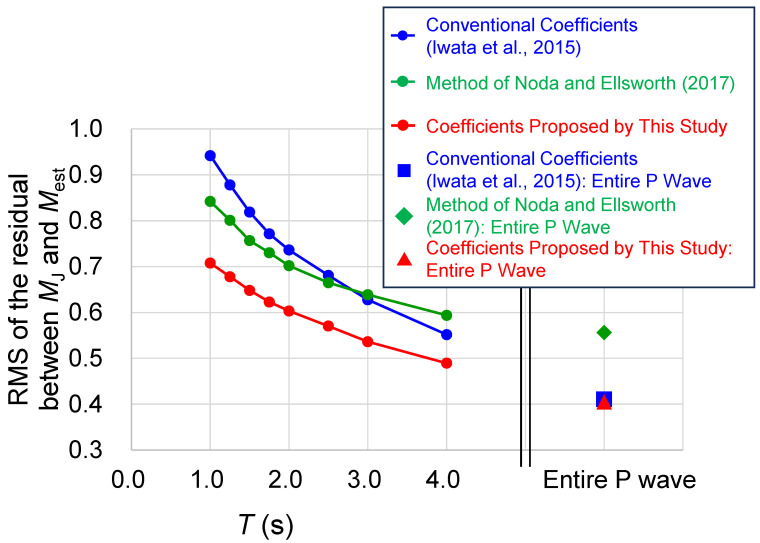
Comparison of the RMS residuals between *M*_J_ and *M*_est_ obtained from each technique. The blue, green and red points indicate the results from the coefficients of Ref. [23] (Iwata et al., 2015), the method of Ref. [24] (Noda and Ellsworth, 2017) and the coefficients proposed in this study, respectively.

**Figure 5 sensors-24-07361-f005:**
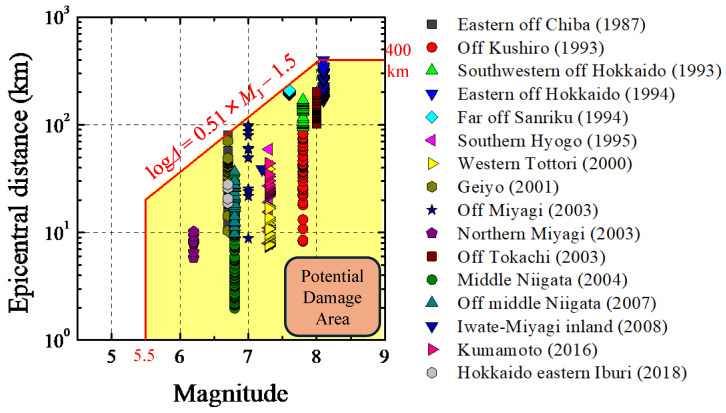
The M-Δ diagram [8,9,10]. The horizontal and vertical axes represent magnitude (JMA magnitude, *M*_J_) and epicentral distance (*Δ*), respectively.

**Figure 6 sensors-24-07361-f006:**
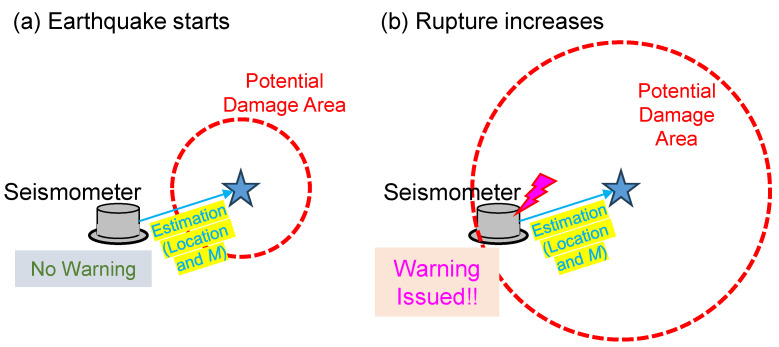
Schematic illustration of the warning output in our investigation: (**a**) at the initial stage of the rupture, no warning is required, because the seismometer estimating the epicenter and *M* is located outside the potential damage area; (**b**) as the rupture progresses, a warning is issued because the seismometer is within the area.

**Figure 7 sensors-24-07361-f007:**
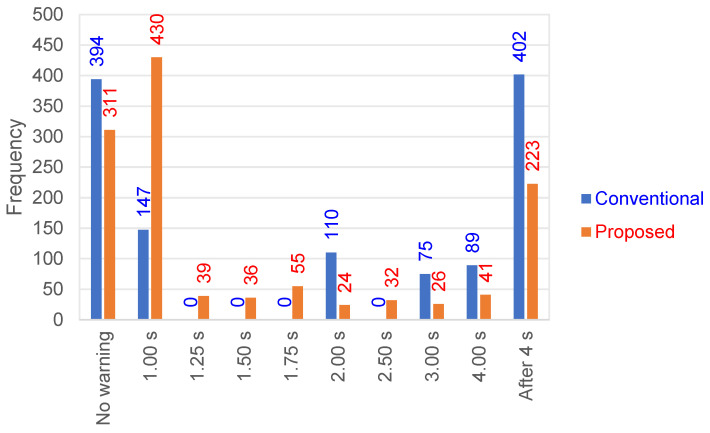
Timing of warning outputs based on the seismic parameters estimated from the conventional and proposed coefficients.

**Table 1 sensors-24-07361-t001:** Coefficients for Equation (1) used in the conventional Shinkansen EEW algorithm [23] and for Equation (2) determined in this study.

*T* (s): Time from the P Onset	Parameters for Equation (1): Used in the Conventional Shinkansen EEW Algorithm		Parameters for Equation (2): Proposed in This Study	
	Pm1	Pm3	Pm1	Pm3[*T*]
1.00	0.9684	6.0015	0.9837	6.6789
1.25				6.6141
1.50				6.5550
1.75				6.5089
2.00				6.4752
2.50				6.4237
3.00				6.3769
4.00				6.3041
Entire P-wave (after 4.00 s)				6.1202

The same values are employed for Pm2 (0.4535) and Pm4 (0.003789).

## Data Availability

The accelerogram data used in this study were collected and distributed by the National Research Institute for Earth Science and Disaster Resilience (http://www.kyoshin.bosai.go.jp/kyoshin; last accessed on 20 September 2024).

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
