# Peer review of "Improving the Rapidity of Magnitude Estimation for Earthquake Early Warning Systems for Railways"

_sensors, 2024, doi:10.3390/s24227361_

Round 1
Reviewer 1 Report
Comments and Suggestions for Authors
Here are my comments:
The manuscript presents a novel approach to improving earthquake early warning (EEW) systems for railways by using time-dependent P-wave displacement analysis to improve the speed and accuracy of magnitude estimation.
The methodology should be more detailed in some sections. In particular, the process of selecting and filtering the data set (noise discrimination) should be expanded upon. In addition, although the coefficients are clearly presented, more explanation of how they were derived or fine-tuned for this specific study would improve reliability for other researchers.
The discussion is largely focused on the Shinkansen system in Japan, which is appropriate given the context. However, it would strengthen the manuscript to briefly discuss whether these improvements could be generalised or adapted for other EEW systems beyond the Japanese rail network.
In your conclusions, you could discuss future research directions. How could this method be adapted to other infrastructure systems beyond railways? In addition, exploring potential extensions, such as integrating other forms of data or extending the analysis to different regions, could be valuable to readers.
Additional technical insights:
More insight into the computational efficiency of the proposed method would be helpful. Does this approach require significantly more computational resources than conventional methods, or is it equally feasible for real-time application?
These improvements could increase the impact of your research and make the results more accessible to a wider audience. Overall, this is a valuable contribution to the field and these refinements will make the manuscript even stronger.
Comments on the Quality of English LanguageAlthough the manuscript is generally clear, some sentences could be simplified to improve readability. For example, there are a few areas where the sentence structure is slightly awkward, leading to potential confusion.
Author Response
Comments 1:
The manuscript presents a novel approach to improving earthquake early warning (EEW) systems for railways by using time-dependent P-wave displacement analysis to improve the speed and accuracy of magnitude estimation.
Response 1:
Thank you very much for your insightful comments and suggestions. We have carefully reviewed the manuscript to the best of our ability and incorporated your feedback. Below, you will find our responses to each of your comments.
Comments 2:
The methodology should be more detailed in some sections. In particular, the process of selecting and filtering the data set (noise discrimination) should be expanded upon. In addition, although the coefficients are clearly presented, more explanation of how they were derived or fine-tuned for this specific study would improve reliability for other researchers.
Response 2:
We have expanded the Data section to clarify the dataset selection and noise discrimination process as follows:
“Smaller or deeper earthquakes, which are less likely to cause damage, are excluded from this study, as the proposed approach is intended for use in EEW. The data with epicentral distances (Δ) less than 200 km are selected. We deem it necessary to utilize records from such distant stations, as many damaging earthquakes occur offshore in Japan” and “To avoid the influence of non-seismic noise in analyzing the data, we visually inspected the waveforms and manually picked P- and S-wave onsets”.
Additionally, we have updated Section 4.1 to explain the derivation of the coefficients:
“Using the least squares analysis with MJ and the peak logDis of the entire P wave, we obtain a value of Pm1 = 0.9837. The role of Pm1 is to adjust the correlation between P-wave amplitude and magnitude. This correlation starts out weaker immediately after the P-wave onset but strengthens over time, reaching its highest level when using the peak amplitude of the entire P wave. We can improve the accuracy of magnitude estimation by determining and fixing Pm1 from data at the point of strongest correlation [24]”.
Comments 3:
The discussion is largely focused on the Shinkansen system in Japan, which is appropriate given the context. However, it would strengthen the manuscript to briefly discuss whether these improvements could be generalised or adapted for other EEW systems beyond the Japanese rail network.
In your conclusions, you could discuss future research directions. How could this method be adapted to other infrastructure systems beyond railways? In addition, exploring potential extensions, such as integrating other forms of data or extending the analysis to different regions, could be valuable to readers.
Response 3:
We have added to the end of the Discussion the following text to broaden the implications of our findings:
“In this article, we focused on applying the proposed method within the railway field in Japan and discussed its effectiveness. However, this approach is anticipated to benefit not only railway applications but also EEW systems for public and in other regions. Moving forward, further studies in this direction are expected to enhance the overall impact of EEW”.
Comments 4:
More insight into the computational efficiency of the proposed method would be helpful. Does this approach require significantly more computational resources than conventional methods, or is it equally feasible for real-time application?
These improvements could increase the impact of your research and make the results more accessible to a wider audience. Overall, this is a valuable contribution to the field and these refinements will make the manuscript even stronger.
Response 4:
We have incorporated also at the end of the Discussion:
“As previously mentioned, the displacement approach allows for updating the M estimate in response to changes in amplitude over time. Since the method proposed in this study adjusts the coefficients of the estimation formula used in M calculation, it has the advantage of not increasing computational costs, making it more feasible for implementation”.
Comments 5:
Although the manuscript is generally clear, some sentences could be simplified to improve readability. For example, there are a few areas where the sentence structure is slightly awkward, leading to potential confusion.
Response 5:
We have reviewed the entire manuscript based on your comments and made all necessary revisions to the best of our ability.
Reviewer 2 Report
Comments and Suggestions for Authors
The paper presents a well-defined approach to improving the rapidity of earthquake magnitude estimation by build a new relationship between Magnitude and P-wave displacement, which is essential for the safety of Shinkansen. However,the author also mentioned it will leads to an increase in the frequency of alerts due to the trade-off between immediacy and accuracy.
In general, the data sources, experimental results and conclusions used in the article are relatively reliable. The English writing needs to be polished a little bit to make the expression clearer.
I think the author needs to explain clearly in two aspects: 1. Regarding equations (1) and (2), the explanation is not clear, for example, "Pm3 represents the intercept of the equation", what does that mean? Pm3(T) is a function of T, what is its specific expression? And how is it defined? 2. In line 124-125, the author needs to give the derivation process and explain how to derive the various coefficients using the mentioned 111,68 acceleration data.
Comments on the Quality of English LanguageThe quality of English in the article is generally clear and well-structured, but there are areas where it could be refined for enhanced readability and precision. Here are some comments for improvement:
-
Technical Clarity: some sentences are overly complex, which can hinder reader comprehension. Simplifying sentence structure, especially in technical explanations (e.g., in equations or method descriptions), could improve flow. For example,Line 15-17 can break into two sentences for clarity, “Our approach reduces the underestimation of M in the initial stages of the P-wave, compared to the conventional technique. As a result, it significantly increases the number of early warnings in the Japanese railway EEW system.” Please check similar problems all through.
-
Consistency in Terminology: Maintaining consistency in technical terminology throughout the article would reduce potential confusion. For example,defining abbreviations like "EEW" only once, rather than repeating the full term in later sections, would also improve the text’s flow.
-
Passive vs. Active Voice: The article predominantly uses passive voice, which is typical for scientific writing but can make sentences feel lengthy. Occasionally switching to active voice, especially when describing the authors' actions (e.g., "We demonstrate" or "Our study shows"), could make the text more engaging.
-
Suggestions for Rewording: Sentences in the introduction and methodology sections could be rephrased to be more concise. For example, “The proposed method inevitably leads to a higher occurrence of warnings” could be simplified to “The proposed method results in more frequent warnings.” This would make the text more direct.
Author Response
Comments 1:
The paper presents a well-defined approach to improving the rapidity of earthquake magnitude estimation by build a new relationship between Magnitude and P-wave displacement, which is essential for the safety of Shinkansen. However, the author also mentioned it will leads to an increase in the frequency of alerts due to the trade-off between immediacy and accuracy.
In general, the data sources, experimental results and conclusions used in the article are relatively reliable. The English writing needs to be polished a little bit to make the expression clearer.
Response 1:
Thank you very much for your valuable comments and suggestions. We have thoroughly reviewed the manuscript and addressed your feedback as fully as possible. Below, we provide our responses to each of your comments.
Comments 2:
I think the author needs to explain clearly in two aspects:
- Regarding equations (1) and (2), the explanation is not clear, for example, "Pm3 represents the intercept of the equation", what does that mean? Pm3(T) is a function of T, what is its specific expression? And how is it defined?
Response 2:
We have addressed your question in the Method section as follows:
“Note that Pm3[T] does not represent a specific mathematical expression; rather, it indicates that the value varies over time and is determined by the data”.
Comments 3:
- In line 124-125, the author needs to give the derivation process and explain how to derive the various coefficients using the mentioned 111,68 acceleration data.
Response 3:
We have clarified this point in Section 4.1:
“Using the least squares analysis with MJ and the peak logDis of the entire P wave, we obtain a value of Pm1 = 0.9837. The role of Pm1 is to adjust the correlation between P-wave amplitude and magnitude. This correlation starts out weaker immediately after the P-wave onset but strengthens over time, reaching its highest level when using the peak amplitude of the entire P wave. We can improve the accuracy of magnitude estimation by determining and fixing Pm1 from data at the point of strongest correlation [24]”.
Comments 4:
The quality of English in the article is generally clear and well-structured, but there are areas where it could be refined for enhanced readability and precision. Here are some comments for improvement:
- Technical Clarity: some sentences are overly complex, which can hinder reader comprehension. Simplifying sentence structure, especially in technical explanations (e.g., in equations or method descriptions), could improve flow. For example,Line 15-17 can break into two sentences for clarity, “Our approach reduces the underestimation of M in the initial stages of the P-wave, compared to the conventional technique. As a result, it significantly increases the number of early warnings in the Japanese railway EEW system.” Please check similar problems all through.
- Consistency in Terminology: Maintaining consistency in technical terminology throughout the article would reduce potential confusion. For example,defining abbreviations like "EEW" only once, rather than repeating the full term in later sections, would also improve the text’s flow.
- Passive vs. Active Voice: The article predominantly uses passive voice, which is typical for scientific writing but can make sentences feel lengthy. Occasionally switching to active voice, especially when describing the authors' actions (e.g., "We demonstrate" or "Our study shows"), could make the text more engaging.
- Suggestions for Rewording: Sentences in the introduction and methodology sections could be rephrased to be more concise. For example, “The proposed method inevitably leads to a higher occurrence of warnings” could be simplified to “The proposed method results in more frequent warnings.” This would make the text more direct.
Response 4:
Based on your feedback, we thoroughly reviewed the entire manuscript and made all necessary revisions as comprehensively as possible.
Reviewer 3 Report
Comments and Suggestions for Authors
The paper considers the problem of declaring a short-term alarm to stop the operation of the high-speed railway network in Japan based on the readings of a network of seismometers recording ground motion after seismic events. A modification of the currently existing decision rules for declaring such an alarm is proposed. The decision rules are based on the use of regression formulas between the earthquake magnitude, epicentral distance, and ground displacement. The traditional formula uses constant coefficients determined by the least squares method based on the information contained in the recorded waveform from the moment of P-wave arrival to the moment of S-wave arrival. The paper proposes a modification based on the dependence of the free term of the regression formula on the time elapsed from the moment of P-wave arrival. A sequence of 8 time moments from the minimum of 1 sec to the maximum of 4 sec with a step of 0.25 sec is considered. It is shown that the introduction of a time dependence in the regression formula leads to a decrease in the underestimation of the magnitude at small time intervals from the P-wave arrival. Thus, the proposed modified decision rule is more cautious, which obviously may lead to an increase in false alarms.
Apparently, the authors have a data set of the traditional train stopping decision rule. If so, it would be interesting to obtain and compare estimates of the probabilities of false alarms and missed targets for the traditional and proposed decision rules.
Author Response
Comments 1:
The paper considers the problem of declaring a short-term alarm to stop the operation of the high-speed railway network in Japan based on the readings of a network of seismometers recording ground motion after seismic events. A modification of the currently existing decision rules for declaring such an alarm is proposed. The decision rules are based on the use of regression formulas between the earthquake magnitude, epicentral distance, and ground displacement. The traditional formula uses constant coefficients determined by the least squares method based on the information contained in the recorded waveform from the moment of P-wave arrival to the moment of S-wave arrival. The paper proposes a modification based on the dependence of the free term of the regression formula on the time elapsed from the moment of P-wave arrival. A sequence of 8 time moments from the minimum of 1 sec to the maximum of 4 sec with a step of 0.25 sec is considered. It is shown that the introduction of a time dependence in the regression formula leads to a decrease in the underestimation of the magnitude at small time intervals from the P-wave arrival. Thus, the proposed modified decision rule is more cautious, which obviously may lead to an increase in false alarms.
Apparently, the authors have a data set of the traditional train stopping decision rule. If so, it would be interesting to obtain and compare estimates of the probabilities of false alarms and missed targets for the traditional and proposed decision rules.
Response 1:
We appreciate your thoughtful summary and insights on the approach of our method.
As described in the main text, the decision rule is to determine whether the seismic station locations fall within the potential damage area on the M-Δ diagram. When a halt signal is issued from seismometers located within this area, substations cut off the power supply, and running trains activate their emergency brakes.
Although this study analyzes waveforms recorded by K-NET rather than those from the Shinkansen EEW system, the decision rule used in this analysis aligns with those in the actual system. Consequently, the frequency of warning outputs shows an approximately 1.5-fold increase when comparing the proposed technique with the conventional method. It may be also interesting for you to note that missed alarms are reduced by approximately 21% using the proposed method compared to the conventional one, as discussed in the Discussion section.
Round 2
Reviewer 1 Report
Comments and Suggestions for Authors
Dear Authors,
Thank you for the thoughtful revisions to the manuscript. The current version reflects significant improvements based on feedback from the initial review. The revised manuscript is comprehensive, clear, and makes a significant contribution to the field of earthquake early warning systems. The detailed explanations, expanded sections, and transparent data processing enhance the credibility and potential impact of your research. With these additional minor refinements, the paper will be even more robust.